# Understanding the Effect of Multiple Domain Deletion in DNA Polymerase I from *Geobacillus* Sp. Strain SK72

**Waqiyuddin Hilmi Hadrawi [1], Anas Norazman [1], Fairolniza Mohd Shariff [1,2], Mohd Shukuri Mohamad Ali [1,3] and Raja Noor Zaliha Raja Abd Rahman [1,2,*]** 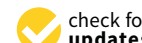

[1]  Enzyme and Microbial Technology Research Centre, Faculty of Biotechnology and Biomolecular Sciences, Universiti Putra Malaysia, UPM Serdang 43400, Selangor Darul Ehsan, Malaysia; waqiyuddinhilmihadrawi@yahoo.com.my (W.H.H.); anasnorazman90@gmail.com (A.N.); fairolniza@upm.edu.my (F.M.S.); mshukuri@upm.edu.my (M.S.M.A.)

[2]  Department of Microbiology, Faculty of Biotechnology and Biomolecular Sciences, Universiti Putra Malaysia, UPM Serdang 43400, Selangor Darul Ehsan, Malaysia

[3]  Department of Biochemistry, Faculty of Biotechnology and Biomolecular Sciences, Universiti Putra Malaysia, UPM Serdang 43400, Selangor Darul Ehsan, Malaysia

*   Correspondence: rnzaliha@upm.edu.my

**Abstract:** The molecular structure of DNA polymerase I or family A polymerases is made up of three major domains that consist of a single polymerase domain with two extra exonuclease domains. When the N-terminal was deleted, the enzyme was still able to perform basic polymerase activity with additional traits that used isothermal amplification. However, the 3′-5′ exonuclease domain that carries a proofreading activity was disabled. Yet, the structure remained attached to the 5′-3′ polymerization domain without affecting its ability. The purpose of this non-functional domain still remains scarce. It either gives negative effects or provides structural support to the DNA polymerase. Here, we compared the effect of deleting each domain against the polymerase activity. The recombinant wild type and its variants were successfully purified and characterized. Interestingly, SK72-Exo (a large fragment excluding the 5′-3′ exonuclease domain) exhibited better catalytic activity than the native SK72 (with all three domains) at similar optimum temperature and pH profile, and it showed longer stability at 70 °C. Meanwhile, SK72-Exo2 (polymerization domain without both the 5′-3′ and 3′-5′ exonuclease domain) displayed the lowest activity with an optimum at 40 °C and favored a more neutral environment. It was also the least stable among the variants, with almost no activity at 50 °C for the first 10 min. In conclusion, cutting both exonuclease domains in DNA polymerase I has a detrimental effect on the polymerization activity and structural stability.

**Keywords:** SK72 DNA polymerase I; polymerase domain; 5′-3′ exonuclease domain; 3′-5′ exonuclease domain; proofreading activity

---

## 1. Introduction

Deoxyribonucleic acid (DNA) polymerase plays a key role in maintaining the genetics information of an organism from one generation to another through the catalysis of double-stranded DNA [1]. This enzyme plays a significant role in the DNA repair pathway to secure the integrity of genetic materials by proofreading and recorrecting any mismatched nucleotide [2,3]. In general, DNA polymerase enzymes have been categorized into seven different families termed as A, B, C, D, X, Y, and RT according to their similarity in amino acid sequences as well as elucidated structure analyses [4,5]. It is noteworthy that family A polymerases are widely studied due to their interesting



structure, which mimics the right hand in humans, and their recent applications in isothermal amplification [6,7].

Family A polymerases, referred to as DNA polymerase I, have three major domains that work as a single chain protein, namely: (1) the 5′-3′ polymerase domain, (2) the 5′-3′ exonuclease domain, and (3) the 3′-5′ exonuclease domain (absent in some enzymes) [8]. The crystal structure of family A polymerases reveal their structural organization and the division of their conserved 5′-3′ polymerase domain into three subdomains called the palm, finger, and thumb. The palm domain plays a role in catalyzing the transfer of phosphoryl groups in the phosphoryl transfer reaction that catalyzes by a two-metal-ion mechanism. The finger domain appears to function in the binding of nucleoside triphosphates with the template base. The thumb domain plays a potential role in the processivity, translocation, and positioning of the DNA. Meanwhile, the 5′-3′ exonuclease domain serves as filling for the gaps present in the Okazaki fragment at the lagging strand, and the 3′-5′ exonuclease domain plays a role in DNA excision repair processes [9–11].

In 1997, the large fragment of DNA polymerase I from *Geobacillus stearothermophilus* (previously *Bacillus stearothermophilus*) namely *Bst*-DNA polymerase was successfully characterized and crystallized [12,13]. *Bst*-DNA polymerase is widely used in loop-mediated isothermal amplification (LAMP) due to its thermolabile characteristic (optimum at 60 °C) and inherent strand displacement activity [14]. The *Bst*-DNA polymerase fragment showed high similarity with the Klenow fragment from *E. coli* that comprised only two major domains (3′-5′ exonuclease and 5′-3′ polymerase domains). This shows that the *Bst*-DNA polymerase is still able to retain its enzymatic activity even by cutting one of the major domains, the 5′-3′ exonuclease domain. The 3′-5′ exonuclease domain in *Bst*-DNA polymerase that carries a proofreading activity is expected to be non-functional due to the lack of conserved residue, which is responsible for the catalytic reaction [15–17]. It is further confirmed by understanding the absence of divalent metal ion interaction among the exonuclease site with the DNA template [13]. Yet, the structure remains attached with the 5′-3′ polymerase domain without affecting its polymerization activity.

The effect of truncation on enzymes, either a few amino acid residues or the major domain, is often investigated and compared to its wild type, which results in improved biological function and structural properties. For instance, a study by Kamaruddin et al. observed an improved crystallizability and anti-aggregation property of AT2 lipase from *Staphylococcus epidermis* AT2 with the removal of only four residues on the C-terminal region [18]. Meanwhile, Lamers et al. reported a DNA polymerase III with an additional domain called Polymerase and Histidinol Phosphatase (PHP). However, the function of this domain is still not entirely understood, as it presumably serves as a proofreading subunit or is likely to be catalytically inactive in some organisms. Thus, deletion of the 60 N-terminal residue showed a loss of activity and suggested that the PHP domain was responsible for maintaining the structural integrity [19,20]. Aother study on the domain structure in polymerase also concluded that the existence of a tethered domain in rat DNA polymerase β influenced the stability and folding of the structure as well as the functional and regulatory properties [21].

In this present work, our aim was to understand the effect of the truncation of multiple domains in SK72 DNA polymerase I. Until now, a comparative study between the full-length DNA polymerase I from *Geobacillus* sp. and its Klenow-like fragments has been poorly understood. Moreover, the sole purpose of a non-functional 3′-5′ exonuclease domain still remains unknown; either it only provides structural support or it damages the function of the polymerase activity. Hence, concerted efforts into understanding the effect of multiple domain deletion in DNA polymerase I is necessary and merits scientific attention. Here, DNA polymerase I (SK72 DNA polymerase I) from *Geobacillus* sp. SK72 isolated from Sungai Kelah hot spring in Perak, Malaysia was used. Several polymerase recombinants were designed where each had a varied number of domains in existence. The polymerase recombinants were further characterized based on their effect toward the biochemical and biophysical properties. The data obtained could provide insights into the effect of major domain deletion in SK72 DNA polymerase I as well as the whole family A polymerase.

## 2. Results and Discussion

### 2.1. Conserved Domain and Structure Analyses

The SK72 DNA polymerase I gene was encoded for 878 polypeptides with an approximate molecular weight and isoelectric point (pI) of 100 kDa and 5.53, respectively. The N-terminal containing the 5′-3′ exonuclease domain was located at residues 1 to 298. The central region of the SK72 DNA polymerase I was made up of a 3′-5′ exonuclease domain (residue 299 to 469), and the C-terminal bearing 5′-3′ polymerase domain started at residue 470 to 878 (Figure 1). Overall, the amino acids consisted of 15.8% negatively charged (Asp + Glu) and 13.4% positively charged (Arg + Lys) residues at neutral pH. The percentage of total charged residues (Arg, Lys, His, Asp, and Glu) in the enzyme was 30.3%. Moreover, the hydrophobic amino acid residues (Ala, Phe, Ile, Leu, Met, Pro, Val, and Trp) occupied 44.6% of total SK72 DNA polymerase I amino acids.

Overall, SK72 DNA polymerase I shares a similar structural framework with DNA polymerase I, consisting three major domains (2 exonucleases and 1 polymerase domains). These domains were identified against three online databases, namely Conserved Domain Search (CDS), InterPro Scan, and Protein Families (Pfam). The domains were further analyzed by predicting their secondary structure by Position-Specific Iterative Basic Local Alignment Search Tool (PSI-BLAST) based secondary structure prediction (PSIPRED) and coupled with comparative modeling software called YASARA to verify the domain boundaries and suitable initial residue for each variant. Absence of the full-length structure of DNA polymerase I from *Geobacillus* sp.; *Taq* polymerase (1TAQ), served as the most reliable template for SK72 DNA polymerase I in the secondary structure prediction and homology modeling, despite having low identity (42%). Meanwhile, Klenow-like polymerase from *G. stearothermophilus* (3TAN) was used for other variants (SK72-Exo and SK72-Exo2) with 99% sequence identity. This is to identify the suitable starting residue and prevent any unwanted cutting of non-important structure that might interfere with the folding and stability of the structure. Most of the α-helix and β-strand structures were preserved, as they were indicators of protein stability [22]. Thus, the codon at the coil structure was used as the initial codon for each variant.

**Figure 1.** Amino acid sequence of the SK72 DNA polymerase I. Note: 5′-3′ exonuclease domain (blue region), 3′-5′ exonuclease domain (red region), and 5′-3′ polymerase domain (yellow region).

Figure 2 shows that all the predicted models were able to maintain the structural integrity of DNA polymerase I, especially the polymerase catalytic region. Although the 3′-5′ exonuclease domain was located near the polymerization domain, the deletion of both exonuclease domains did not disrupt the overall shape of the polymerization domain entirely. The full-length structure was dominated

by the α-helical structure with 51.2%, followed by 10.7% and 38.1% of β-strand and other structures, respectively, while the number of helix and strand structures in SK72-Exo and SK72-Exo2 were reduced prior to the elimination of the domain. All structures displayed an open-state formation where the thumb and finger subdomain were distant from each other and exposed the catalytic site located at the palm subdomain, while allowing the access of the DNA template [23,24]. A Mg$^{2+}$ ion was predicted to react with the polymerase active site at the position of Tyr 654, Asp 653, and Asp 830, which acted as a cofactor and initiated the nucleotide addition [25,26]. The overall quality of all the models falls between 95% and 98%, which is within that of a reliable and good folded protein.

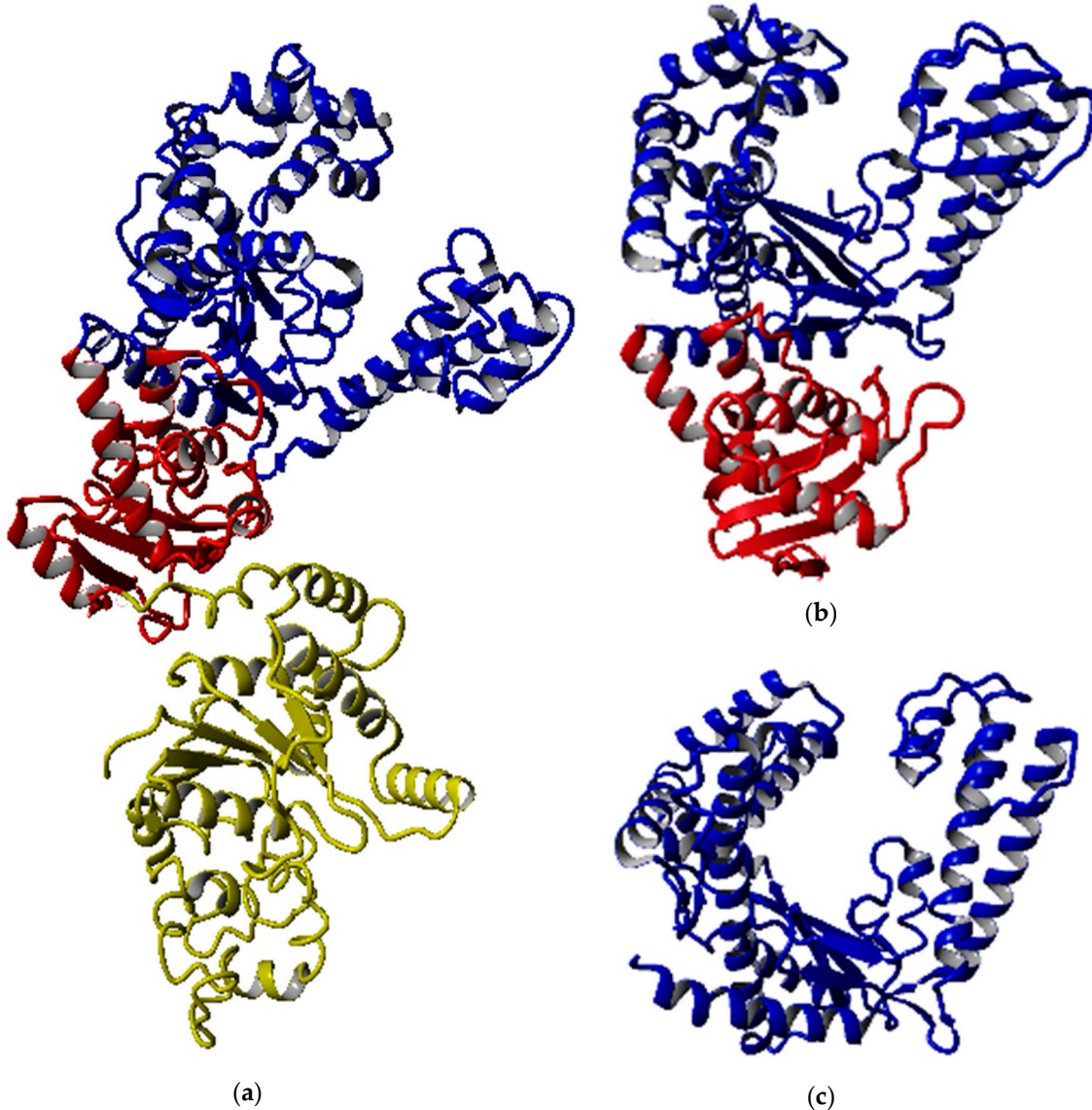

(a)　　　　　　　　　　　　　　　　　　　　　　(c)

**Figure 2.** Modeled structures of SK72 DNA polymerase. (**a**) SK72 DNA polymerase, (**b**) SK72-Exo, and (**c**) SK72-Exo2. Note: 5'-3' polymerase domain (blue), 3'-5' exonuclease domain (red), and 5'-3' exonuclease domain (yellow).

### 2.2. Cloning, Overexpression, and Purification of SK72 DNA Polymerase I and Its Variants

Three recombinant constructs were designed, namely: (1) SK72, SK72 DNA polymerase I containing all three major domains; (2) SK72-Exo, SK72 DNA polymerase I with a deleted N-terminus 5'-3' exonuclease domain, and (3) SK72-Exo2, SK72 DNA polymerase I with only the polymerase domain. All constructs were fused with a polyhistidine tag on the N-terminus region (Figure 3).

The overexpression for each construct varied. For SK72 and SK72-Exo, the production of proteins was accumulated in the form of soluble parts at 37 °C for 12 h with 0.25 mM of Isopropyl β-d-1-thiogalactopyranoside (IPTG) induction. Meanwhile, the SK72-Exo2 was overexpressed at 16 °C for 12 h with 0.5 mM of IPTG induction, as the protein showed insoluble expression at 37 °C and 25 °C of incubation. This suggests that the expression at low temperature helps to reduce protein aggregations and assist in protein folding, thus decreasing the formation of inclusion bodies [27,28]. Then, the crude enzymes obtained underwent heat treatment at 60 °C for 30 min to reduce some background expression except for SK72-Exo2 due to its protein instability. All proteins were successfully purified via single-step purification using $Ni^{2+}$ Sepharose affinity chromatography with almost 80% of impurities eliminated (Figure 3). The purified proteins were further subjected to the desalting procedure prior to characterization.

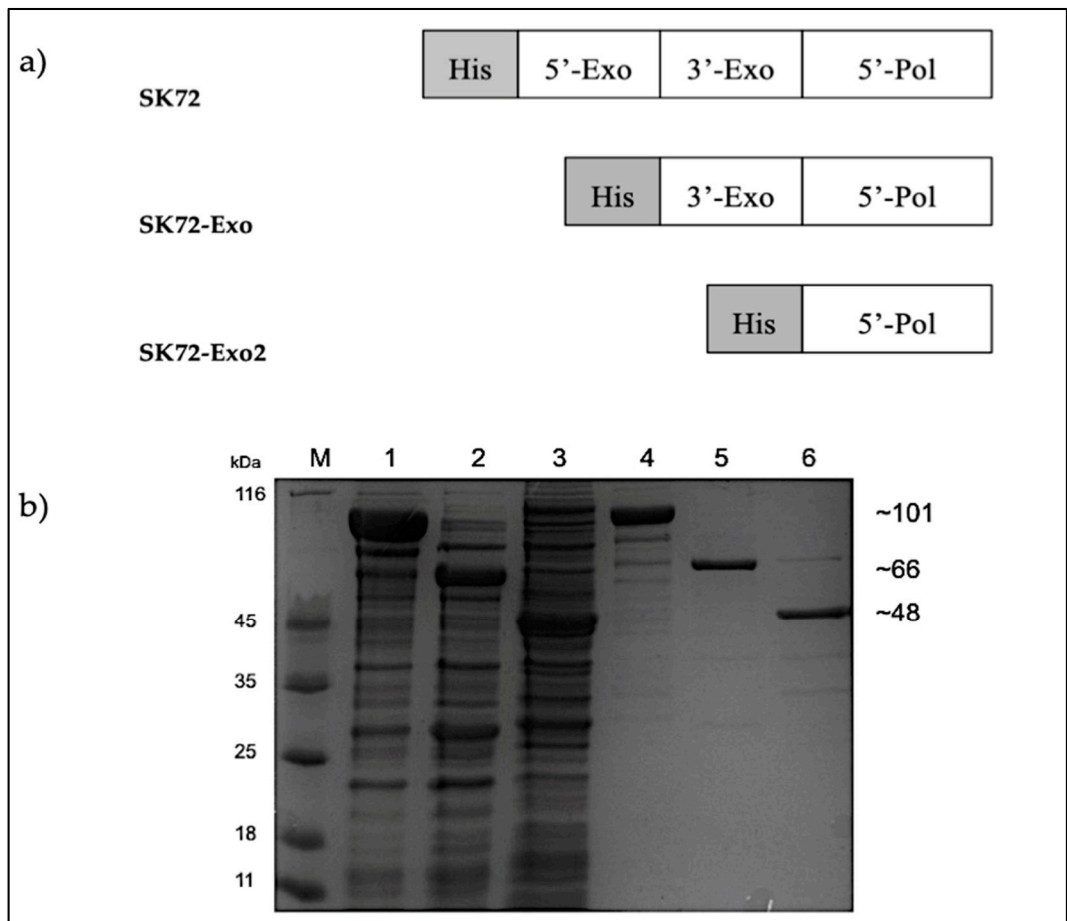

**Figure 3.** The SK72 DNA polymerase I and its variants. (**a**) Schematic representation of SK72 DNA polymerase I and its variants fused with a polyhistidine tag. (**b**) Expression and purification of SK72 DNA polymerase I and its variants. The overexpressed (lane 1 to 3) and purified (lane 4 to 6) of SK72 DNA polymerase I and its variants. Note: SK72 (lanes 1 and 4), SK72-Exo (lanes 2 and 5), SK72-Exo2 (lanes 3 and 6), and M = unstained protein marker (Thermo Fisher, City, Waltham, MA, USA).

*2.3. Characterization of SK72 DNA Polymerase I and Its Variants*

2.3.1. Effect of Temperature on SK72 Polymerase Activity and Its Variants

Characteristics of the SK72 DNA polymerase I and its variants were determined using a standard DNA polymerase assay with a final concentration of 2 µg/µL [29]. The effect of temperature on SK72 DNA polymerase activity was studied by measuring the activity from 20 to 80 °C with an interval of 10 °C. In comparison with all three variants, SK72-Exo demonstrated the highest polymerase

activity profile, followed by SK72 and finally SK72-Exo2 with 72.1 U/mL, 48.3 U/mL, and 21.8 U/mL, respectively (Figure 4). Interestingly, both SK72 and SK72-Exo showed similar optimum temperature activity at 60 °C but dropped at 70 °C. This shows that the deletion of the 5'-3' exonuclease domain had no effect on the temperature profile but enhanced the polymerase activity, suggesting that the N-terminal domain deletion promoted better flexibility and rigidity [30]. Similar phenomena were also monitored with the alkaline α-amylase Amy703, where the effect of deletion of its N-terminal domain (N-Amy) caused an increase of optimum temperature from 40 to 50 °C [31,32]. Meanwhile, the SK72-Exo2 achieved its optimum activity at 40 °C, which is slightly lower compared to SK72 and SK72-Exo. Obviously, deleting both exonucleases brought a major effect on the core catalytic site; it lowered the optimum activity and shifted the temperature profile 20 °C to the left. At 50 °C, the activity rapidly inactivated with a total inhibition at higher temperature that might disrupt the structural integrity of SK72-Exo2. This characteristic is similar to the DNA polymerase I produced by *Psychrobacillus* sp., which is a marine psychrophilic bacterium that is able to perform DNA synthesis at ambient temperature [8].

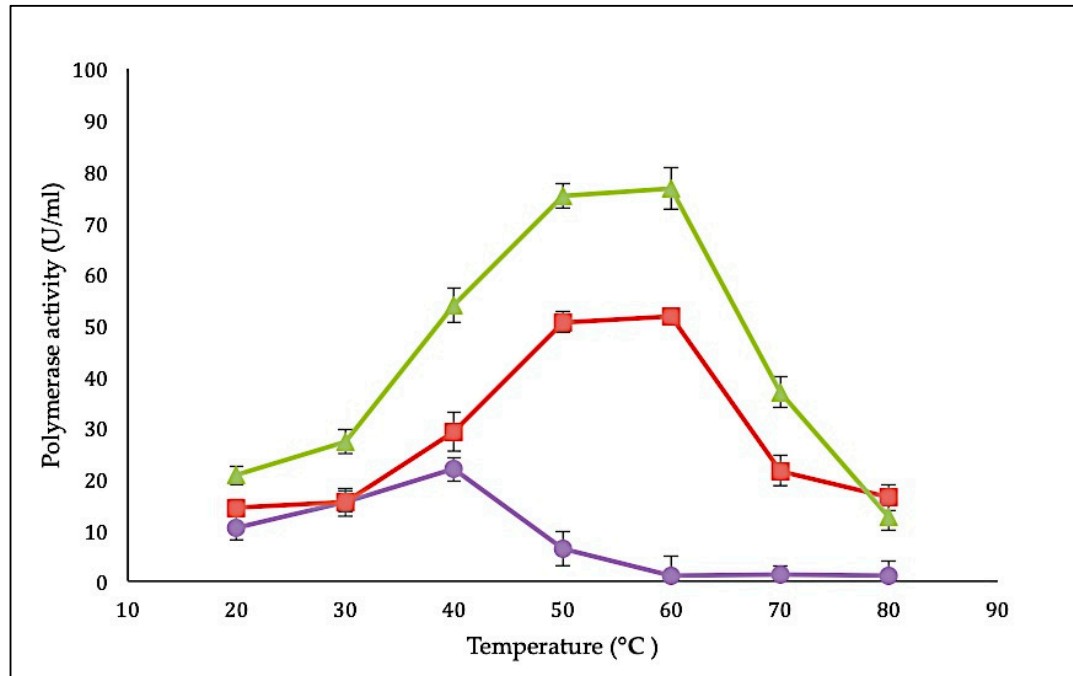

**Figure 4.** Effect of temperature on SK72 DNA polymerase activity and its variants. Optimal temperature was determined by varying the assay temperatures. Symbols represent the enzyme variants; SK72 (filled square), SK72-Exo (filled triangle), and SK72-Exo2 (filled circle). Error bars represent standard deviation (*n* = 3) The absence of the bar indicates that the error is smaller than the symbols.

### 2.3.2. Effect of pH on SK72 Polymerase Activity and Its Variants

The effect of pH on purified SK72 DNA polymerase and its variants was tested on different pH ranging from 4 to 11. The full-length SK72 polymerase showed maximum activity at pH 9 similar to that of SK72-Exo, suggesting that both enzymes preferred alkaline environment to work. SK72-Exo continued to show improvement by obtaining the highest catalytic activity and displayed a broader pH profile with higher activity at pH 8 to 9. Wang et al. reported that a superoxide dismutase (SOD) from *Bacillus subtilis* contained an extra 244 residue on the N-terminal domain (NTD) exhibiting a slight effect on the pH profile upon the NTD deletion that changes from pH 4 to pH 5 [33]. Meanwhile, the effect of truncation of the GST-C domain in AMS3 lipase from *Pseudomonas* sp. also promotes similar optimum pH activity with its native enzyme at pH 8 [30]. This showed that removing the 5'-3'

exonuclease domain does not greatly impact the overall structure and function of the polymerase, since neither major changes occurred in the pH profile.

In contrast, the SK72-Exo2 pH profile was shifted to pH 7 and decreased gradually starting from pH 8 to pH 11 (Figure 5). The effect was that it tremendously changed the ionic environment, which might have been caused by the reduction of the high number of positive and negatively charged residues and the localization of the exonuclease domain. Among the 469 residues that were removed, 72 residues carried a positive charge (26 arginine, 37 lysine, and 9 histidine), while 79 residues carried a negative charge (30 aspartic acid and 49 glutamic acid). These residues contribute to the pKa of the enzyme, and the deletion of this high number of important residues may greatly impact the ionic interaction [30]. As compared to SK72-Exo, the 3′-5′ exonuclease domain was located toward the polymerase active site separated by a linker region that was covalently attached with the polymerase domain. Thus, deleting this domain structure may influence the overall structure and dynamics of the enzyme [21]. A study by Du et al. reported that the truncation of the N-terminal domain of an invertase (uninv2) shifted the optimal pH from pH 4.5 to 6.0, which was caused by the changes on the pKa values and electrostatic potential of the ionizable groups on the active sites. Even though the N-terminal domain does not carry any enzymatic activity, it may disturb the pKas of the catalytic residues [34,35]. The absence of the N-terminal exonuclease strongly improved the characteristic of SK72 DNA polymerase I. Meanwhile, deletion of the 3′-5′ exonuclease domain had a severe effect on the ionic interfaces of the enzyme.

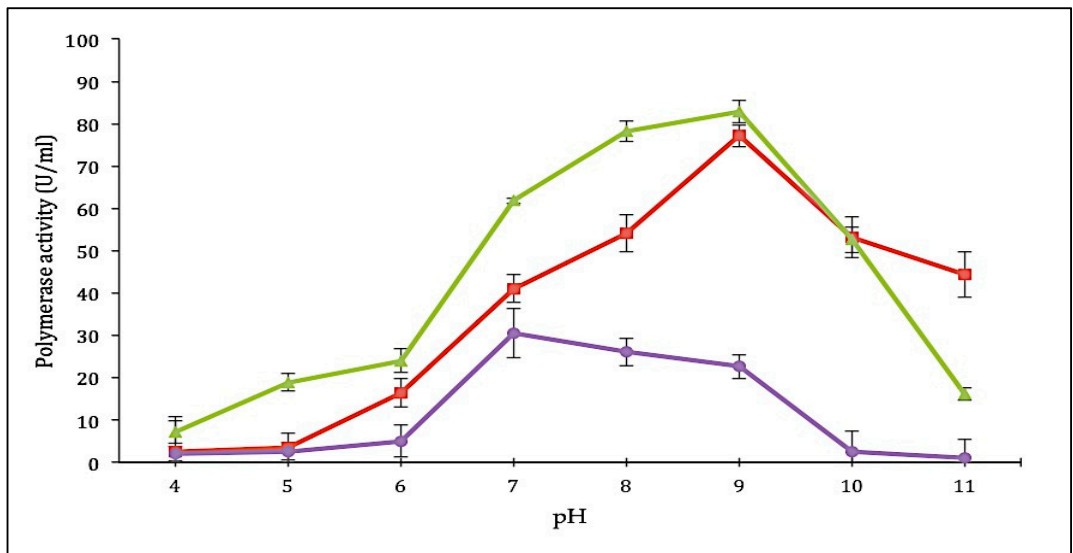

**Figure 5.** pH profile of SK72 DNA polymerase and its variants. Symbols represent the enzyme variants; SK72 (filled square), SK72-Exo (filled triangle), and SK72-Exo2 (filled circle). The optimal pH was determined by measuring the enzyme activity in different buffer systems ranging from pH 4 to 11. The error bars represent standard deviation (*n* = 3). The absence of the bar indicates that the error is smaller than the symbols.

2.3.3. Effect of MgCl$_2$ Concentration on SK72 Polymerase Activity and Its Variants

Specific metal ions are often involved in enhancing the catalytic activity of an enzyme. Magnesium (Mg$^{2+}$) ions play a critical role in facilitating the polymerization reaction. In this study, the influence of Mg$^{2+}$ ions on polymerase activity was tested at various concentrations ranging from 0 to 5 mM. Polymerase activity was determined relative to control conditions in the absence of any metal ions (Figure 6). For SK72, the polymerase activity was stimulated maximally at 3 mM, similar to SK72-Exo2. Both enzymes also showed decreased activity when supplied with higher MgCl$_2$ concentration. Unlike the full-length form, SK72-Exo showed a broader concentration range, with a maximum at 4 mM. More than 60% of activity was observed when supplemented with Mg$^{2+}$ as low as 1 mM, but it

showed a reduction of 20% of its activity at 5 mM. The Stoffel fragment of Taq polymerase also needed a higher concentration of $Mg^{2+}$ for optimal activity compared to the full-length Taq polymerase I [36,37]. RKOD DNA polymerase also exhibited high optimum activity at 2 mM [38]. This demonstrates that a minimum concentration of 3 mM was sufficient to obtain the maximum enzyme activity. Conversely, without any addition of $MgCl_2$ into the reaction, the polymerase activity was greatly impaired with almost no activity, suggesting that a divalent metal ion was needed in every polymerization reaction. This shows that the removal of both extra domains in DNA polymerase I does not give any detrimental effect on the binding of metal ion toward the catalytic site.

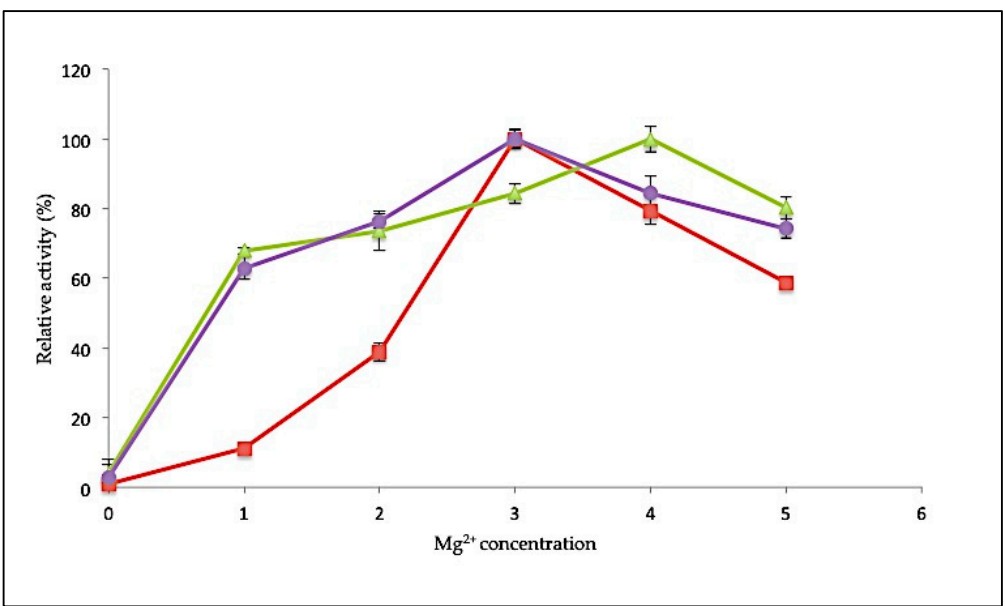

**Figure 6.** Effect of magnesium ion concentration toward SK72 DNA polymerase I activity and its variants. Note: SK72 (red), SK72-Exo (green), and SK72-Exo2 (purple). The assay took place at optimum temperature for each variant and incubation for 30 min. The control consists of an absence of metal ions (0 mM) relative to the maximal activity for each experiment. Error bars represent standard deviation ($n = 3$). Absence of the bar indicates that the error is smaller than the symbols.

### 2.3.4. Thermostability of SK72 Polymerase Activity and Its Variants

The SK72 DNA polymerase I and its variants were earlier identified as a slightly thermostable enzyme that was remarkably active at temperatures ranging from 40 to 60 °C. Apart from the effect of N-terminal domain deletion on the temperature and pH profile, this large fragment of SK72 DNA polymerase I could synergistically increase the protein stability and rigidity. A strong conformational structure is a key factor in enzyme stabilization. On this relevance, Figure 7 denotes the thermostability study of SK72 DNA polymerase as compared to its variants at three different temperatures: 50, 60, and 70 °C. Excellent stability was observed in SK72-Exo, where it was able to maintain almost 80% of its activity throughout the total incubation period at 50 and 60 °C but decreasing at 70 °C after 20 min of incubation. Meanwhile, SK72 was only able to retain half of its activity up to 60 min both at 50 and 60 °C and dropping drastically after 10 min of incubation at 70 °C.

In another related study, the truncated 5′-nuclease domain in *Taq* polymerase also showed higher thermostability compared to its full length. It is hypothesized that the 5′-nuclease domain might be considered as a flexible domain [36,39]. Deletion of the flexible domain has enhanced the protein stability due to reducing the number of unfolded conformations, which was also proven in SK72-Exo [40]. In the case of SK72-Exo2, it demonstrated a loss of activity for all temperatures within the first 10 min of incubation, as the optimum temperature profile previously also fell sharply at 50 °C. Thermostability in enzymes is often linked with the presence of the disulfide bond formed by

cysteine residues [41,42]. Based on the amino sequence analysis, deletion of the 5′-3′ exonuclease domain did not remove the existing disulfide bridge based on the native form. Meanwhile, deleting the 3′-nuclease domain removed one cysteine residue on the sequence, thus impairing the disulfide bond formation. This shows that absence of the 3′-nuclease domain promotes an extensive effect on the structure integrity and rigidity of SK72 DNA polymerase I.

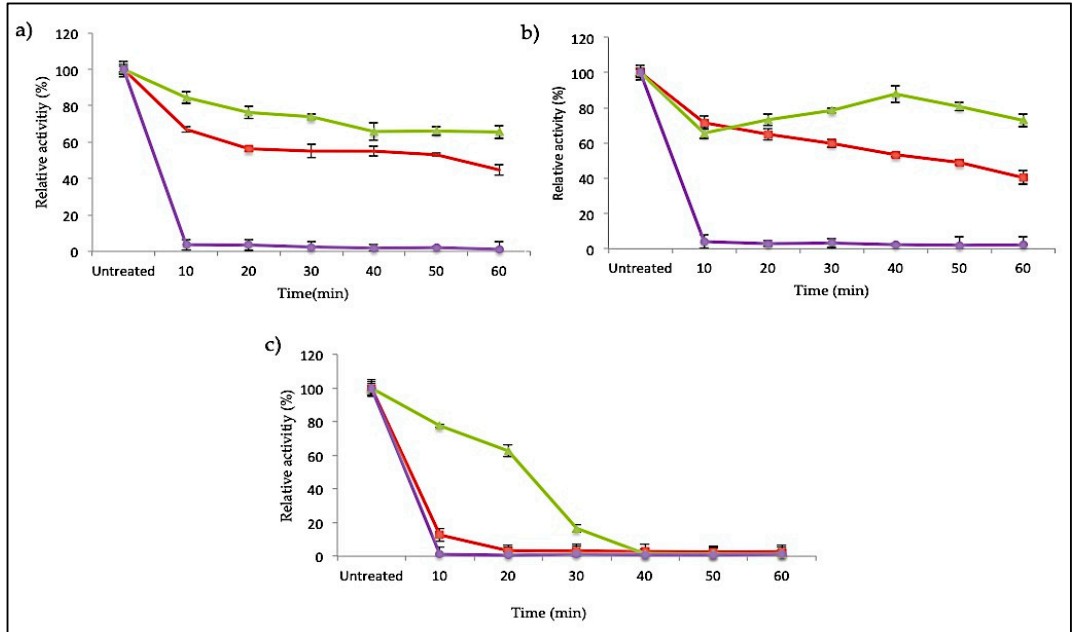

**Figure 7.** Thermal stability profile of SK72 DNA polymerase and its variants. (**a**) 50 °C, (**b**) 60 °C, and (**c**) 70 °C. Purified enzymes were pre-incubated for 30 min at different temperatures prior to polymerase assay at the optimum temperature of each variant. Untreated enzymes were measured relative to the activity of the enzymes without pre-incubation. Symbols represent the enzyme variants; SK72 (filled triangle), SK72-Exo (filled square), and SK72-Exo2 (filled circle). Error bars represent standard deviation (*n* = 3). The absence of the bar indicates that the error is smaller than the symbols.

## 2.4. Thermal Denaturation Analysis of SK72 DNA Polymerase I and Its Variants

Further thermal stability study was conducted using a JASCO J810-spectrapolarimeter (JASCO, Tokyo, Japan). Measurement was conducted at 222 nm for temperature ranging from 20 to 90 °C. Generally, soluble globular proteins will undergo structural rearrangement as the temperature environment rises. Throughout the increasing amount of heat applied, the folded protein structure will start to deteriorate until it reaches its melting temperature (Tm). In these studies, the midpoint between the folded and unfolded state of the SK72 DNA polymerase I and its variants was obtained (Figure 8. For both SK72 and SK72-Exo, the $T_m$ values were 69.84 °C and 71.01 °C, respectively. The full-length *Taq* polymerase and its Stoffel fragment exhibit a similar half-life at 97.5 °C, but the Stoffel fragment promotes longer resistance [37]. Meanwhile, the SK72-Exo2 thermal denaturation profile was adjusted to the left with $T_m$ of 46.22 °C. The effect of removing the central 3′-5′ exonuclease domain tremendously changed the overall structure stability of the polymerase, resulting in a lower melting point. This also in correlation with the thermostability result that previously showed an activity drop at 50 °C, which may be caused by the loss of disulfide bridge formation. The location of the first Cys390 was at the 3′-exonuclease domain, while the second Cys847 was located at the 5′-polymerase domain. Thus, deleting the 3′-exonuclease domain disrupts the only disulfide bond that is connected by these residues, resulting in a decrease of the melting point. A similar result was also obtained upon the mutation within the PHP domain in Pol III, suggesting that a decrease in $T_m$ values highlights that the non-catalytic domain was structurally integrated with the rest of the polymerase as

one unit [20]. Thus, this major thermal shift suggested that the 3′-5′ exonuclease domain does bring a structural support to the overall structure and stability of the protein.

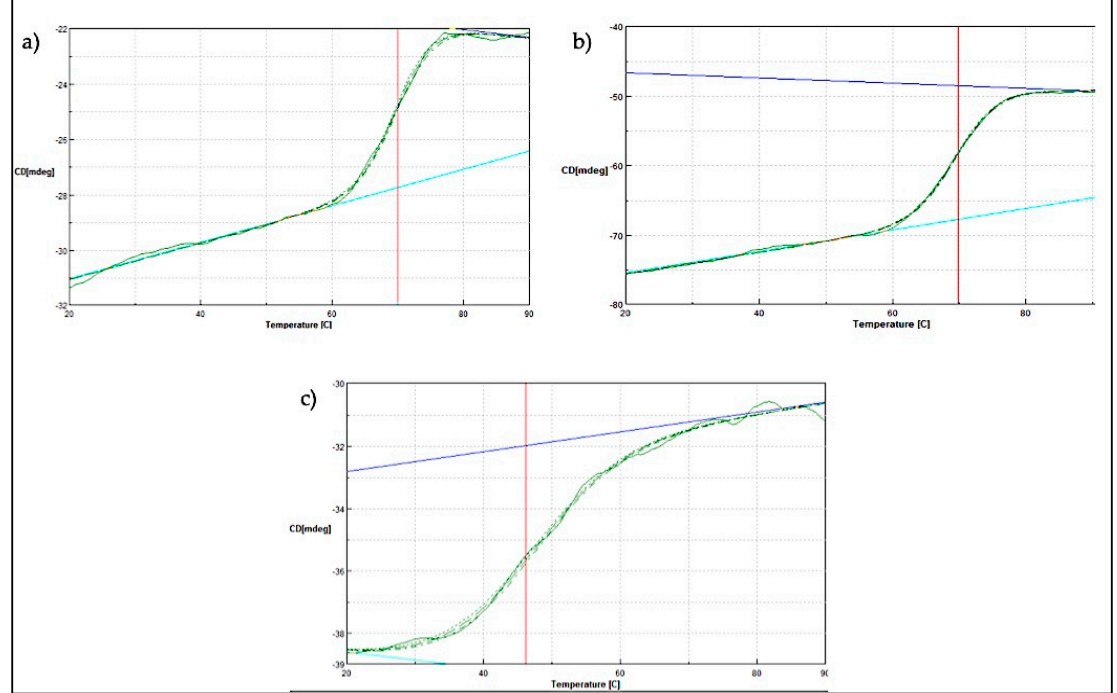

**Figure 8.** Thermal denaturation profile of SK72 DNA polymerase and its variants. (**a**) SK72, (**b**) SK72-Exo, (**c**) SK72-Exo2. Vertical line (red) indicating the melting point value of each variant when tested from temperatures ranging from 20 to 90 °C.

*2.5. Secondary Structure Analysis of SK72 DNA Polymerase I and Its Variants*

Circular dichroism (CD) measurements have been widely used to follow the equilibrium between helical structures and unordered conformations. The CD spectra (molecular ellipticity) of SK72 DNA polymerase I was analyzed between 190 and 260 nm at 25 °C. Wavelengths at 220 and 190 nm were set to monitor the transition of α-helix and β-strand structures, respectively, as they exhibited characteristic signals at this wavelength. A summary of the secondary structure estimation is presented in Table 1. The stability of the protein structure is often linked to its overall helical and strand content. Here, the percentage of α-helix took most of the variant's structure with SK72-Exo being the highest, followed by SK72 and SK72-Exo2. The effect of truncation also decreased the amount of strand structure gradually, as can be seen in Figure 2. The overall structure of SK72 DNA polymerase was mainly made up by a helical structure that covered the β-sheet where the catalytic region resided. As expected, the high percentage of α-helix content enhanced the stability and activity of SK72-Exo. Meanwhile, the removal of the 3′-exonuclease domain decreased the protein stability due to loss of the protein secondary structure, followed by an increased in the disordered conformation.

**Table 1.** Secondary structure content determination.

|  | Amount of Secondary Structure (%) | | |
|---|---|---|---|
|  | **SK72** | **SK72-Exo** | **SK72-Exo2** |
| Helix | 47.0 | 49.1 | 45.0 |
| Sheet | 18.5 | 12.9 | 5.5 |
| Turn | 11.4 | 17.8 | 19.6 |
| Coil | 23.1 | 20.3 | 29.9 |

## 3. Materials and Methods

### 3.1. Strains, Vectors, and Growth Medium

The *Geobacillus* sp. strain SK72 used in this research was obtained from Enzyme and Microbial Technology Research Centre, Faculty of Biotechnology and Biomolecular Sciences, Universiti Putra Malaysia. The bacterium was previously isolated from the Sungai Kelah hot spring in Perak, Malaysia. *Escherichia coli* strains TOP10 and BL21 (DE3) were purchased from Novagen (Merck, Darmstadt, Germany). The pGEM-T Easy was purchased from Promega (Madison, Wisconsin, USA), and expression vector pET28a was obtained from Enzyme and Microbial Technology Research Centre, Faculty of Biotechnology and Biomolecular Sciences, Universiti Putra Malaysia. Nutrient agar and broth media were purchased from Merck (Darmstadt, Germany). *E. coli* Luria-Bertani (LB) agar and broth growth media were purchased from Merck (Germany).

### 3.2. Isolation of SK72 DNA Polymerase I Gene

Genomic DNA of *Geobacillus* sp. SK72 was isolated according to a standard procedure by Qiagen (Hilden, Germany). DNA polymerase I genes of seven (7) related species including *Anoxybacillus* sp. NB, *Geobacillus stereothermophilus*, *Geobacillus thermodenitrificans*, *Geobacillus caldoxylosilyticu*, *Bacillus caldotenax*, *Thermus thermophiles* strain HJ6, and *Thermus filiformis* RT41A obtained from the National Centre for Biotechnology Information (NCBI) (www.ncbi.nlm.nih.gov/blast/) were used to perform multiple sequences alignments and were selected for degenerate primers design.

The primers termed as F-degenerate (GAYTAYTCGCAAATYGARYTG) and R-degenerate (GAGYTCGTCATGSACYTG) were used to PCR amplify the gene fragment on a thermocycler (Bio-Rad, Hercules, CA, USA) in a 20 µL reaction mixture containing 100 ng (1 µL) of template DNA, 0.25 µM (0.5 µL) each of forward and reverse primers, 2 µL of 10X*Taq* polymerase buffer, 1.5 mM (1.2 µL) of MgCl$_2$, 25 U (0.10 µL) of *Taq* DNA polymerase, and 14.2 µL of distilled water. The PCR incubation conditions were set to have an initial denaturation temperature at 95 °C for 3 min, followed by 25 cycles at 95 °C for 1 min, 50.0 °C for 1 min, and 68.0 °C for 2 min and 15s, followed by a final extension at 68.0 °C for 10 min. The PCR product was analyzed on 1.0% (*w/v*) agarose gel. The amplified fragment showed high identity to certain regions of DNA polymerase I genes in *Geobacillus kaustohilus* (BA000043.1) (100%), *Bacillus* sp. G (EF198253.1) (99%), and *Geobacillus kaue* strain E1 (FJ215761.1) (88%).

### 3.3. Cloning of SK72 DNA Polymerase I

Subsequently, F-SK72 pol (ATGAGATTGAAAAAAAAGCTTGTTTTA) and R-SK72 pol (TTATTTCGCGTCATACCATGTCGAG) primers were designed from the flanking regions of the *Geobacillus kaue* strain E1 (FJ215761.1) DNA polymerase I gene and used for another round of PCR with the same incubation conditions mentioned above to amplify the SK72 pol. All the gene fragments obtained during the cloning steps were purified by gel extraction (Qiagen, Hilden, Germany) and directly sequenced before or after ligation into pGEM-T Easy (Promega, Madison, WI, USA).

### 3.4. Nucleotide Sequencing and Amino Acid Analysis

All PCR products and successful ligated products were sequenced at First Base Sdn Bhd (Apical Scientific, Selangor, Malaysia). Sequence analysis was performed using Basic Local Alignment Tools (BLAST) at the National Centre for Biotechnology Information (NCBI) (www.ncbi.nlm.nih.gov/blast/). Multiple sequences alignments were done using the CLUSTAL W Programme [43]. A full nucleotide (accession no.: MG190359.1) and amino acid (accession no.: AXU98681.1) sequence were submitted into the GenBank database.

### 3.5. Domains Analysis of SK72 DNA Polymerase and Its Variants

The conserved domains were identified using the Conserved Domain Search (CDS), InterPro Scan, and Protein Families (Pfam) database. Then, the secondary structure prediction was performed using online tools—namely, PSI-blast based secondary structure prediction (PSIPRED), coupled with comparative modeling software called YASARA to confirmed the domain boundaries and initial codon for each variants. Both tools used the closest structure for each variant against the Protein Data Bank (PDB) database. Then, all predicted models were evaluated for the reliability of the structure using four automated servers, namely ERRAT [44], Verify 3D [45], QMEAN [46], and Ramachandran plot developed by RAMPAGE (http://www-cryst.bioc.cam.ac.uk/).

### 3.6. Construction of SK72 Pol and Its Variants

**SK72**. A full-length gene of the SK72 DNA polymerase I was PCR amplified from the synthesized vector, pUCIDT, using the SK72-F1 and SK72-R1 primers listed in Table 2. The underlined sequences are restriction sites for *XbaI* and *SalI* enzymes in the forward and reverse primers, respectively. The amplified products (2.6 kb) were purified using a gel extraction kit (Qiagen, Germany), treated with *NdeI* and *SalI* restriction enzymes, ligated into a pET28a expression vector, and transformed into *E. coli* BL21 (DE3). Positive transformants were screened on an LB agar plate incorporated with 50 µg/mL of kanamycin at 37 °C overnight. The recombinant plasmid designated pET28/SK72 was validated by colony PCR, enzymes digestion, and sequencing.

**Table 2.** List of primers used for subcloning into a pET28 expression vector.

| Primer Name | Sequence (5′-3′) | Restriction Site |
|---|---|---|
| SK72-F1 | ATCCCATATGATGCGCCTGAAAAAAAAACTGGTGCTGATTG | NdeI |
| SK72-R1 | TGCGTCGACTCATTTGGCATCATACCAAGTAC | SalI |
| SK72-Exo-F2 | GAAACATATGGCCAAAATGGCGTTTACGCTGGC | NdeI |
| SK72-Exo2-F3 | GCTTCATATGAATGAACAGGATCGGCTGCTGGTG | NdeI |

**SK72-Exo**. The partial N-terminus SK72 DNA polymerase I, known as a large fragment, encodes for the 3′-5′ exonuclease and 5′-3′ polymerization domain amplified from the pUCIDT vector using SK72-Exo-F2/SK72-R1 primers. The resultant 1.75-kbp DNA fragments and pET28 were digested with *NdeI* and *SalI*, ligated, and transformed into *E. coli* BL21 (DE3) according to the standard protocol. The recombinant plasmid designated as pET28/SK72-Exo was validated by colony PCR, enzymes digestion, and sequencing.

**SK72-Exo2**. The partial C-terminus SK72 DNA polymerase I (deleted both 5′-3′ and 3′-5′ exonuclease) nucleotide sequence was amplified from the pUCIDT vector using SK72-Exo2-F3/SK72-R1 primers. The resulting 1.2-kbp DNA fragments and pET28 were digested with *NdeI* and *SalI*, followed by ligation, and transformed into *E. coli* BL21 (DE3) according to the standard protocol. The obtained recombinant plasmid was purified, digested, and further validated by sequencing. The resultant plasmid was named pET28/SK72-Exo.

### 3.7. Expression and Purification of SK72 Polymerase I and Its Variants

Successfully transformed colonies of *E. coli* BL21 (DE3) that carried the recombinant expression vector pET28/SK72 were cultured in 200 mL of LB broth in a 1 L conical flask in the presence of kanamycin to an optical density (OD) of 0.5. The overexpressed *E. coli* cells were induced using IPTG to a final concentration according to the constructs. The cells were harvested by centrifugation ($8000\times g$, 20 min) after 12 h of induction, and the pellet was suspended in 10 mL of phosphate buffer (pH 7.4). The cells were lysed by sonication, and the insoluble fraction was removed by centrifugation ($4000\times g$, 20 min). Purification was performed using $Ni^{2+}$-Sepharose affinity chromatography aided by a AKTA Protein Purification System (GE, Chicago, IL, USA). The soluble fraction of the cell lysates was equilibrated with binding buffer (pH 7.4, 30 mM imidazole) and loaded into a 5 mL HisTrap FF

(GE, Chicago, IL, USA). The unbound protein was washed away from the column by the binding buffer at a flow rate of 1 mL/min, and the bound protein was eluted with elution buffer (pH 7.4, 500 mM imidazole) in a gradient imidazole concentration. The eluted fractions were collected and analyzed by SDS-PAGE on 10% gel.

*3.8. Polymerase Activity Assay*

Characterization of the SK72 DNA polymerase activity and its variants were performed by incorporating a radiolabeled nucleotide during DNA polymerization synthesis [29]. A reaction mixture (50 µL) containing 25 mM Tris-Cl, 4 mM MgCl$_2$, 25 µM deoxynucleoside triphosphate (dNTP), 0.5 µCi thymidine triphosphate (TTP)[methyl-H$^3$], 0.2 mg/mL bovine serum albumin (BSA), 0.1% Tween 20, 5 µg activated calf thymus DNA, and 100 µg of SK72 DNA polymerase was prepared. The enzyme assay was performed at 60 °C for 30 min and spotted the mixture on a glass fiber filter disc. The filter was air dried, washed twice with 0.5 M sodium phosphate pH 7.0, and dried with 70% ethanol. The filter was placed into a scintillation vial, and 2 mL of scintillation fluid was added into the vial. The radioactivity of the incorporated nucleotides was measured using a Packard Liquid Scintillation Counter (Perkin Elmer, Waltham, MA, USA). One unit of polymerase activity is defined as the rate of incorporation of 10 pmol of the labeled dNTP into the DNA at 60 °C in 30 min.

*3.9. Thermal Stability, Optimal Temperature and pH, and MgCl$_2$ Concentration Analyses*

For thermal stability, purified SK72 DNA polymerase and its variants were incubated at three different temperatures (50, 60, and 70 °C) for one hour. The incubated samples were immediately chilled on ice, and the enzyme activity was determined using the standard polymerase assay conditions described above. The temperature and pH optimum of the enzyme was investigated by measuring the polymerase activity at a range of 20 to 90 °C and from pH 4 to pH 11, respectively based on the protocol described above for the polymerase activity assay. The optimal ion concentration was examined similarly as in the polymerase activity assay, using an MgCl$_2$ concentration in the range of 0 to 5 mM.

*3.10. Secondary Structure and Melting Point Estimation*

The thermal stability and the secondary structure estimation of the enzymes were analyzed by heating in circular dichroism. For secondary structure estimation, the reactions were conducted in 500 µL containing 0.2 mg/mL enzyme and 10 mM sodium phosphate storage buffer. The temperature was set at 20 °C, and the absorbance was read ranging from 260 to 190 nm. The melting point was examined with total reactions containing 5 mL of 2 mg/mL enzyme and 5 mM sodium phosphate storage buffer. The temperature was increased from 20 to 95 °C, with an increment of 1 °C per second, and the absorbance signal was collected from wavelength 280 to 190 nm.

**4. Conclusions**

The present work was to compare the effect of the truncation of multiple domains in SK72 DNA polymerase I. The native SK72 and its variants were successfully expressed and purified via single-step affinity chromatography. The removal of the 5'-3' exonuclease domain (SK72-Exo) was found to improve the enzyme activity but maintain the temperature and pH profile similar to the native SK72 at 60 °C and pH 9, respectively. SK72-Exo promoted better thermostability compared to the two other variants; it was able to retain almost 80% of its activity at 50 and 60 °C for an hour, and decreasing at 70 °C after 20 min. Deletion of the flexible domain (5'-3' exonuclease), which contains a high number of α-helix structures, may contribute to the enhancement of enzyme activity and stability. Meanwhile, the effect of deletion on both N-terminal domains (SK72-Exo2) caused an extensive effect toward the polymerase catalytic activity that collapsed the overall polymerase structure and function. Thus, we conclude that the 3'-5' exonuclease domain is considered as a major structural domain instead of carrying any catalytic activity. This is also linked to the location of the 3'-5' exonuclease domain that

was very close, which covalently linked to the polymerase domain. The improvements of SK72-Exo in catalytic activity and thermostability were also seen between the Stoffel fragments in *Taq* DNA polymerase I and its wild type [37], but there were none in *Geobacillus sp*. In addition, the effect of cutting both exonucleases diminished the core catalytic domain that displayed similar results to those found in the *E. coli* DNA polymerase III upon the PHP domain deletion [20]. This is the first report that distinguishes the characteristics of the large fragment of *Bst* DNA polymerase I from *Geobacillus sp*. against its full-length form. Thus, a simple engineering technique was performed that provides insights into the effect of major domain deletion in SK72 DNA polymerase as well as family A polymerases.

**Author Contributions:** Conceptualization, W.H.H. and R.N.Z.R.A.R.; methodology, W.H.H., A.N., and R.N.Z.R.A.R.; validation, R.N.Z.R.A.R., M.S.M.A. and F.M.S.; formal analysis, W.H.H., A.N., and R.N.Z.R.A.R.; investigation, W.H.H.; Resources, R.N.Z.R.A.R., M.S.M.A. and F.M.S.; data curation, W.H.H. and R.N.Z.R.A.R.; writing (review and editing) W.H.H., and R.N.Z.R.A.R.; visualization, W.H.H. and R.N.Z.R.A.R.; supervision, R.N.Z.R.A.R., M.S.M.A. and F.M.S.; project administration, R.N.Z.R.A.R.; funding acquisition, R.N.Z.R.A.R., M.S.M.A. and F.M.S. All authors have read and agreed to the published version of the manuscript.

**Funding:** Putra Grant funded this research, (GP-IPS/2017/9600800).

**Acknowledgments:** W.H.H. was supported by Graduate Research Fellowship (GRF) fund and research grant (GP-IPS/2017/9600800) from Universiti Putra Malaysia.

**Conflicts of Interest:** The authors declare no conflict of interest.

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
