# Peer review of "Understanding the Effect of Multiple Domain Deletion in DNA Polymerase I from Geobacillus Sp. Strain SK72"

_catalysts, doi:10.3390/catal10080936_

Round 1

Reviewer 1 Report

The authors report an interesting study to delineate the functional domains in SK72 DNA polymerase I. They describe in detail various properties of the truncation mutant derivatives of the enzyme. I congratulate the authors for their work and would recommend this article be accepted after addressing the following comments:

Section 2.1: The authors conclude that all predicted models of mutant proteins were able to maintain the structural integrity of the polymerase, especially in the catalytic region. They do not mention how the predicted models were deduced. A brief description of how this was achieved should be mentioned in the relevant paragraph in this section.

Section 2.3: Were the concentrations of the protein preparations equalized before determining the activity? If so, please mention.

Section 2.4: Please comment on the effect of loss of the disulphide bridge in the exo2 mutant on enzyme melting temperature.

Section 4: Please comment on the implications of these findings on polymerase enzyme engineering and in the enzymology field in general.

Section 4: Please compare these findings to the already known polymerase domain mutants and discuss these results in the wider context.

Lastly, the manuscript will improve greatly upon a careful copy edit and thorough edit for syntax.

Reviewer 2 Report

In this work Hadrawi et al. investigate the polymerization activity of the A-family polymerase from Geobacillus sp, and specifically explore the effects of truncation of exonuclease domains  over a range of experimental conditions including temperature and pH. Using a radio-labeld nucleotide assay they find that the deletion of one exonuclease domain (SK72-exo) enhances polymerization activity, while the deletion of two exonuclease domains (SK72-exo2) substantially reduces polymerization activity. They investigate the thermostability of each polymerase truncation and find the SK72-exo2 has substantially reduced thermostability compared to SK72 and SK72-exo. Overall, the article presents a series of interesting results that are well described and helpful in understanding the structure-function of A-family polymerases and therefore will warrant publication after several moderate issues are addressed. 

Specific comments:

1) The English is mostly understandable, however the grammar would benefit from copy-editing. At some points in the article the grammar makes it difficult to follow the arguments beings made. 

2) Line 200-201: The authors write "Characteristics of the SK72 DNA polymerase I were determined using standard DNA polymerase assay." This statement does not reference either a research article or an appropriate appendix. This discussion is included in section 3.8 so a reference to the materials and methods section and/or a reference to the radio-labeled nucleotide assay at this point in the text would be sufficient.

3) Line 241-244: The authors write "Wang et al. reported a superoxide dismutase (SODs) from Bacillus subtilis contains an extra 244 residue on the N-terminal domain (NTD) exhibit slight effect on the pH profile upon the NTD deletion, which still considered both as acidophilic [33]. Meanwhile, the effect of truncation of GST-C domain in AMS3 lipase also promote similar optimum pH activity with its native enzyme at pH 8 [29]."

3.1) I am not sure what point the authors are trying to make here. Is their point that each of these three enzymes from the same organism have optimum activity in the pH 8-9 range? 

3.2) Also, is the use of acidophilic here correct for enzymes that prefer operating in basic pH? 

3.3) I think the authors are getting at some interesting ideas here, but I think they need to be presented much more clearly. I think taking 2-3 more sentences here to add more discussion to these ideas will be very helpful for the reader.

4) Line 256-257: "Study by Du et al., reported that the truncation of the N-terminal domain of an invertase (uninv2) shifted the optimal pH from pH 4.5 to 6.0, which make it more suitable for industrial application [35]."

4.1) This sentence is in the middle of a discussion on the pH-dependence of SK72-exo2, and seems out of place. At the very least, to use the reference in the discussion I would argue that the rather than discussing industrial applications that the authors should instead explicitly discuss how this reference supports their proposition that deletion of large domains of the enzyme can change the optimal pH conditions. 

5) In figure 6, the authors show the data as a bar chart rather than a plot (as in figure 4/5/7). I think this data is better presented if it is stylistically consistent with figure 4, figure 5 and figure 7. 

6) Also in figure 6, I am slightly confused about how the normalization is being done. In line 281 the authors state "Polymerase ctivity (sic) was determined relative to control conditions in the absence of any metal ions (Figure 6)."

6.1) This seems to suggest to me that they are normalizing the data to an experiment with no Magnesium ions, but that clearly isn't the case looking at figure 6. It seems like the normalization done in figure 6 is relative to the maximal activity for each experiment. Can the authors verify this / explicitly state it in the figure caption
